# Partial Pulpotomy in Young Permanent Teeth: A Systematic Review and Meta-Analysis

**DOI:** 10.3390/children10091447

**Published:** 2023-08-24

**Authors:** Nicole Camoni, Maria Grazia Cagetti, Silvia Cirio, Marcella Esteves-Oliveira, Guglielmo Campus

**Affiliations:** 1Department of Biomedical, Surgical and Dental Science, University of Milan, Via Beldiletto 1, 20142 Milan, Italy; n.camoni@gmail.com (N.C.); silvia.cirio@unimi.it (S.C.); 2Department of Restorative Dentistry and Endodontology, Justus-Liebig-University Giessen, 35392 Giessen, Germany; marcella.esteves@dentist.med.uni-giessen.de; 3Department of Restorative, Preventive and Pediatric Dentistry, University of Bern, Freiburgstrasse 7, 3012 Bern, Switzerland; guglielmo.campus@unibe.ch

**Keywords:** pulpotomy, endodontics, dental caries, pediatric dentistry, systematic review

## Abstract

The aim of the present systematic review was to evaluate the clinical and radiographic success of partial pulpotomies in deep caries processes or post-eruptive defects in young, vital permanent teeth. Four electronic databases, PubMed, Scopus, Embase, and Google Scholar, were searched, followed by a manual search in the reference lists. Randomized controlled trials evaluating partial pulpotomy with a follow-up period of ≥12 months were included. A meta-analysis using a random effects model was performed. A total of 3127 articles were retrieved, and after duplicate removal, 2642 were screened by title and abstract; 1 additional article was found during the manual search, and 79 were identified. Finally, six papers were included in the review. Regardless of the partial pulpotomy modalities or material used, there were no significant differences between clinical and radiographic success rates (cumulative success rate 91.8–92.3%). Five studies were included in the meta-analysis that did not indicate any statistically significant differences in success rates when mineral trioxide aggregate was compared to other materials (95% confidence interval: 0.239 to 1.016; *p* = 0.055). The present research systematically evaluates the evidence and summarizes the available data on partial pulpotomy in young permanent teeth. Given its high clinical and radiographic success rate, partial pulpotomy should be considered when vital pulpal therapy needs to be performed in highly damaged young permanent teeth, as this procedure provides a biological benefit and allows more invasive endodontic treatments to be postponed.

## 1. Introduction

Avoiding irreversible dental treatments such as pulpectomy or extractions whenever possible is mandatory, especially in young patients. With the current state of knowledge, the emergence of new materials, and the increased consideration for minimally invasive treatments, partial pulpotomy (PP) is becoming a viable option that clinicians should consider when treating young patients [1,2,3].

Severe caries and enamel defects such as molar incisor hypomineralization (MIH) affect many children worldwide: the prevalence of early childhood caries is around 48%, while it is 15% for MIH [1,2]. First, molars are particularly affected by caries as they are in a very vulnerable state, erupting at 6 years of age when brushing skills are often inadequate [4,5]. With regard to the pulp of teeth with MIH, a histological difference has been found compared to sound teeth: the presence of leucocytes is more significant, and innervation is altered due to the expression of neuropeptides and ion channels, resulting in peripheral sensitization [6]. Due to the hypomineralized enamel, an overlay of a carious process on MIH teeth is frequent, and if undetected, this can rapidly progress to the pulp [7]. Adopting the most conservative and least invasive treatments is necessary to maintain these teeth and all others affected as long as possible [8,9].

Considering this scenario, partial pulpotomy may represent a therapeutic option. The first reference to the term ‘partial pulpotomy’ in the dental literature was made by Cvek in 1978. The author reported that 96% of teeth with crown fractures healed after surgical amputation of 2–3 mm of inflamed pulp tissue [3]. Nowadays, partial pulpotomy is one of the vital pulpal treatments that allows physiological root development in young teeth with an open apex, avoiding root treatment or extraction [9,10]. PP is performed in cases of dental trauma and symptomatic deep caries in the dentin or when the tooth is severely damaged due to an enamel development defect [11].

Partial pulpotomy differs from total pulpotomy because the entire coronal pulp is removed in the latter procedure, and a drug is applied directly to the root canal orifices. It also differs from direct pulp cupping because the exposed pulp is only covered with dental material to facilitate the formation of a protective barrier [12]. In the case of PP, the recommended extent for removal of the inflamed pulp varies from 2 to 4 mm in different works [13]. All authors agree on the need to preserve as much of the pulp rich in coronal cells; this strategy increases the chances of healing due to physiological dentin deposition in the amputated area. Hemostasis, in the case of healthy pulp, should occur within 4–10 min, although different times are reported in the literature; if hemostasis is not achieved, more invasive treatment may be necessary [13,14].

Emerging materials and techniques that provide a good seal and better biocompatibility allow partial pulpotomy to be performed with less uncertainty [15]. Calcium hydroxide (CH) is a widely used alkaline material with a pH of up to 12.5, producing a bactericidal effect and allowing hard tissue formation [16]. However, concerns have been raised about its toxicity due to its caustic effect, inability to adhere to dentin, degradation over time, and tunnel defects in the restorative dentin bridge [17,18]. In recent years, the search for better materials has led mineral trioxide aggregate (MTA) to be considered the gold standard for vital pulp therapies [19,20]. Already 20 years ago, Aeinehci et al. [21] stated that a thicker dentinal bridge and a more evident odontoblastic layer were present in human teeth treated with MTA compared to those treated with CH [21]. Several studies have reported good physical and biocompatibility characteristics of MTA; the material allows satisfactory sealing, optimal marginal adaptation, and maintains a high pH for a long time [22,23]. In addition, it cannot be inhibited by blood or moisture and has low solubility [24].

Another material to consider for partial pulpotomy is Biodentine (BD), a calcium silicate cement (CSC) widely used for vital pulpal therapy [25,26]. The physical properties of BD compared to MTA are a reduced setting time and the replacement of bismuth oxide with zirconium oxide as a radiopacifier [15]. Biodentine has good biocompatibility and bioactivity, homogeneity, sealing ability, and high compressive strength [15]. Furthermore, BD has been reported to cause less coronal discoloration than MTA [27]. Finally, laser therapy could be considered an auxiliary tool for vital pulp therapy, as it can be used to achieve hemostasis and coagulation [28].

For the above reasons and current knowledge, partial pulpotomy is a viable treatment option. However, it is still unclear which of the numerous pulp capping materials provides the best clinical efficacy and tooth survival rate when used for the partial pulpotomy technique.

### Rationale of Study and Objective

The rationale of the current review and meta-analysis is to find in the current literature whether partial pulpotomy, which is a conservative treatment that is still little practiced by dentists, can be, in the presence of correct diagnosis and management, an appropriate treatment in young teeth. Therefore, the present review aimed to systematically evaluate the evidence regarding the clinical and radiographic success and pathological outcomes of the techniques and materials used for partial pulpotomy. The current meta-analysis included studies investigating PP for deep cariogenic processes and post-eruptive defects in young viable permanent teeth. To determine the effectiveness of mineral trioxide aggregate in partial pulpotomy, the following null hypothesis was tested: the success rate of PP with MTA is higher than with other materials, such as Biodentine or calcium hydroxide.

## 2. Materials and Methods

### 2.1. Protocol and Registration

The present review was registered in the International Register of Systematic Reviews (PROSPERO) in March 2023 (CRD42023408988); the drafting follows the Preferred Reporting Items for Systematic Reviews and Meta-Analyses (PRISMA) guidelines (Appendix A).

### 2.2. Eligibility Criteria

The research question was formulated based on PICO and aimed to answer the following question: what is the success rate of partial pulpotomy in vital permanent teeth of children and young adults with severe caries lesions or developmental defects at 12 months? The following inclusion criteria were adopted: studies that included paediatric and young populations (age < 24 years) with vital permanent teeth with caries and/or severe enamel defects treated with a partial pulpotomy. The age threshold of 24 years was chosen because, according to the literature, the cell signalling pathways associated with bacterial inflammation are weaker in the pulp of children and young adults, which could explain the better endodontic results [29,30]. Furthermore, the World Health Organization’s definition of “young” covers the age range of 10 to 24 years [31]. The comparators considered were teeth treated by partial pulpotomy with different materials, MTA, calcium hydroxide, and Biodentine, used alone or in combination with other materials/devices; teeth treated with other pulpal therapies were not considered. No sex or health condition limitations were indicated. Only randomized clinical trials written in English and published between 1 January 2012 and 31 December 2022 were considered. Exclusion criteria were studies that included participants with teeth treated only with other vital pulp therapies, such as complete pulpotomy and direct or indirect pulp capping, traumatized teeth, or partial pulpotomies with a follow-up of less than 12 months.

### 2.3. Search Strategy

The electronic search was conducted using three databases: PubMed (National Library of Medicine), Embase (Elsevier), and Scopus (Elsevier). The search was performed from 15 October to 31 December 2022. A search of the grey literature (Google Scholar) was made in March 2023. All the references were uploaded onto the Endnote^®^ software for duplicate management and study selection. The search strategy for each database is displayed in the Appendix A. Finally, the reference lists of the studies included were hand searched to identify additional records.

### 2.4. Study Selection and Data Extraction

After the exclusion of duplicates, two independent authors (NC and MGC) screened the papers according to title and abstract; in case of doubt, the opinion of a third author (GC) was sought. After full-text evaluation, selected articles were uploaded onto a Microsoft Office Excel^®^ spreadsheet (Appendix A). Data extraction was performed in duplicate (NC and MEO), including a description of the study design, outcome, variables assessed, and results. Every effort was made to obtain the original data from the authors, who were contacted by e-mail and/or ResearchGate^®^. Cohen’s kappa value for reviewer agreement was determined for abstracts and full texts.

### 2.5. Risk of Bias and Quality Evaluation of Included Studies

The risk of bias in the included studies was assessed using the ROB-2 tool [32]. The biases evaluated were confounding, selection of participants, classification of interventions, deviation from intended interventions, missing data, measurement of outcomes, and selection of reported results. The discussion resolved divergences.

### 2.6. Outcome Measures

The primary outcome of the present review was the clinical success rate assessed as the absence of pain, swelling, mobility, abscess, percussion tenderness, and/or the radiographic success rate considered as the absence of periapical radiolucency and periodontal ligament enlargement. Secondary outcomes were the restoration type and oral-health-related quality of life as measured using standardized questionnaires.

### 2.7. Heterogeneity

The heterogeneity was evaluated for correspondence between subjects’ characteristics, interventions, and outcomes. The random effects analysis was applied, and statistical heterogeneity was performed using the I^2^ statistic; values over 50% indicated heterogeneity and significance was present at *p* < 0.1.

### 2.8. Synthesis of the Results

The MedCalc^®^ package was employed for the analysis. A meta-analysis was considered appropriate and conducted in the presence of studies with comparable data, *i.e.*, reporting the same outcome, interventions, and same follow-up time (NC and GC). The total number of treated subjects with events was analyzed and compared for dichotomous data. The confidence interval was calculated at 95%.

## 3. Results

### 3.1. Reliability and Validity

A total of 3127 articles were retrieved, and after duplicate removal, 2642 were screened by title and abstract and 2564 were excluded (Appendix A); 78 papers were identified, and 1 additional article was found during the search in the studies’ reference lists. After full-text evaluation, 73 papers were discharged (Appendix A) and 6 studies were included in the systematic review, of which 5 were used in the meta-analysis [3,33,34,35,36]. The search strategy can be seen in the PRISMA flowchart (Figure 1). The Cohen’s kappa value for the inter-reviewer agreement was 0.58 at the title and abstract screening (95.6% agreement) and 0.83 at full-text screening (96.2% agreement).

### 3.2. Study Characteristics

All included studies were parallel RCTs published between 2014 and 2022 and can be found on PubMed, Embase, and Scopus. The general characteristics of the included studies are reported in Table 1. All the studies were carried out in university/college settings and received university grants except two [3,33].

Table 2 shows the main characteristics of the six studies included. The sample sizes ranged from 50 [6] to 105 [33] patients, with a corresponding number of teeth ranging between 50 [6] and 119 [33]; the age of the participants ranged from 6 [34,36] to 15 [36]. Regarding interventions, all the studies evaluated the success rate of partial pulpotomy at 12 months; the other follow-ups were performed at different time points at 1 [36], 3 [3,6,33,36], 6 [3,6,33,34,35,36], and 24 months [3,6,34]. The most common dental material used for pulp therapy was MTA, while the final restoration was performed with many different techniques: composite resin [34,36], stainless steel crown [6,33,35], glass ionomer cement [6,36], and amalgam [3]. All the studies considered success based on clinical and radiographical outcomes. Clinical failure was declared in case of pain, swelling, mobility, abscess, and tenderness to percussion. All the studies used intraoral radiographies; radiographical failure was instead considered in periapical radiolucency cases and periodontal ligament widening in all studies.

One study, not included in the meta-analysis, compared partial pulpotomy with other viable pulpal therapies [6]: indirect pulpal treatment, partial, cervical, and total pulpotomy. The study was the only one to treat teeth with MIH and showed a higher clinical and radiographic success rate at 24 months with indirect pulp treatment (95.8%), but the results were not statistically significant if compared to the success rate of partial or cervical pulpotomy (85.7% for both). The five studies included in the meta-analysis compared the success rate of partial pulpotomy with MTA (ProRoot MTA^®^) with that of other materials/device: Chailertvanitkul et al. [3] and Ozgur et al. [34] with calcium hydroxide (Dycal^®^ and CH Merk Germany^®^, respectively), Uyar et al. [33] with calcium hydroxide (CH Merck^®^, Germany) and Biodentine^®^, Uesrichai et al. [35] with Biodentine^®^ and Tozar et al. [36] with MTA plus an erbium CrYGG laser (WaterLase iPlus^®^; Biolase, Irvine, CA, USA).

The data in Table 2 were calculated using the proportions for dropouts and overall success; the data were found in the tables of the selected manuscripts.

### 3.3. Risk of Bias Assessment

Figure 2 presents the main results of the risk of bias assessment of the included studies. In three studies [33,35,36], a certain risk of bias was found, while in the others, a low risk of bias was found [3,6,34]. The randomization process for patient allocation and deviations from the intended interventions were judged to be low risk for all selected studies. The studies were rated as at a low risk of attrition bias because dropouts were low. The outcome measurement was clearly stated in most studies, as was information on methods and standardization of procedures.

### 3.4. Qualitative Synthesis

The overall success rate considering clinical and radiographic success of partial pulpotomy at 12-month follow-up was greater than 85% in all studies. In the studies considered for the meta-analysis, partial pulpotomy with calcium hydroxide, Biodentine, and MTA plus laser had a cumulative success rate of 91.8%. In comparison, partial pulpotomy with MTA had a cumulative success rate of 92.3%. Saline or hypochlorite solution did not significantly affect the outcomes [34]. The Er: CrYSGG laser did not show significantly better results when applied before MTA [36].

Concerning secondary outcomes, none of the included studies compared treatments to the amount of pulp removed, type of restoration, or oral-health-related quality of life using standardized questionnaires.

### 3.5. Quantitative Synthesis

Quantitative synthesis was achieved by combining five papers that compared the success rate of partial pulpotomy using MTA compared to other materials such as calcium hydroxide, Biodentine, or MTA + laser Er [3,33,34,35,36]. A random effects model showed no statistical significance in the success rate: *p*-value = 0.7759, 95%CI 0.00; 56.04. The heterogeneity among studies was low, I2 (inconsistency) = 0%. The resulting forest plot is shown in Figure 3.

Statistical analysis of the publication bias was not performed as fewer than 10 studies were included.

## 4. Discussion

The main finding of the present review is that partial pulpectomy has been confirmed to be an effective treatment for deep carious lesions or MIH in young permanent teeth without the material used to dress the pulp affecting its effectiveness, thus rejecting the null hypothesis. To the authors’ knowledge, the present systematic review is the first to have performed a meta-analysis of the available studies on the clinical/radiographic effectiveness of partial pulpotomy using different materials in children and young adults. There are not yet many RCTs on PP, but those identified are of good quality and generally agree that this vital pulp therapy has a high clinical and radiographic success rate, exceeding 85% at a 12-month follow-up. These results agree with those described in another systematic review on partial pulpotomy which found a clinical success rate of 93% at 12 months [41]. An umbrella analysis found overall success rates for complete and partial pulpotomies of 88.5% and 90.6%, respectively [42]. At the same time, viable direct pulp capping techniques appear to have a lower success rate, above 75% at 12 months [12]. As reported, the material used on the amputated pulp does not influence the success rate. The results indicate that all materials examined, i.e., MTA, calcium hydroxide, Biodentine, and MTA + erbium (Er) laser, produced satisfactory clinical performance. However, concerns may arise regarding the relatively short follow-up period (12 months) of the included studies, as treatment stability over time may also depend on the performance of the pulpotomy material used. Therefore, in the state of the art and knowledge, further well-designed randomized clinical trials are needed to assess which material is most effective for long-term partial pulpotomy. Especially when newly erupted teeth are treated, adequate follow-ups are necessary to evaluate the success of therapy and apical closure. In the latter cases, it is essential to preserve pulpal viability as long as possible to increase the longevity of the treated tooth.

Although the actual pulp status can only be determined histologically, a 96.6% correlation has been found between reversible pulpitis’s clinical and histological status [43]. The key to the success of partial pulpotomy lies in the following factors: accurate pre-treatment diagnosis (reversible pulpitis), correct isolation of the tooth, adequate removal of the infected pulp, a reasonable choice of materials, and congruent restoration. Bleeding control is also crucial in the success of PP: hemostasis should be achieved within 2–4 min if proper pulpal removal has been performed [6]. The present review has not identified a restorative material or technique superior in the success of PP; however, in endodontics, the importance of an excellent marginal seal that prevents further invasion of microorganisms is well known [44].

The current review considered studies in which PP had been performed for deep carious lesions (ICDAS 4, 5, 6) and post-eruptive breakdown, as the latter is often superimposed on a carious lesion, such as in cases of severe MIH [37]. The site of exposure of the tooth could influence the success rate [40]; more failures have been found with axial than occlusal exposures. Occlusal exposures have a more favorable prognosis because the more permeable cervical third is preserved, and tooth isolation is easier [40,45]. Including post-eruptive lesions adds value to the present work, as treating this type of lesion is a daily activity for many pediatric dentists worldwide. Due to the scarcity of selected studies, it was impossible to compare the success of partial pulpotomy performed in decayed teeth or with MIH. However, this could be a target for future studies, as knowing which technique or material works best would be vital in improving clinical outcomes of this very prevalent enamel development defect.

Further studies on pulp treatments for MIH-affected teeth are needed, especially now that classification systems such as the MIH severity scoring system (MIH-SSS) are becoming popular among pediatric dentists [38]. Awareness of the MIH-SSS should be disseminated among general dentists, as it could be assumed that many MIH teeth are included in clinical research considered as decayed teeth. Indeed, studies confirm the lack of knowledge of the developmental defects of enamel among general dentists [39,46,47].

The American Association of Endodontics and the European Society of Endodontology guidelines support vital pulp techniques, even on teeth with closed apex or mature pulp [48,49]. The new clinical trend is to avoid or at least delay invasive treatments, as the possibility of maintaining part of the pulp vital now that materials and techniques are highly specialized is important; dissemination of the vital pulp therapies among clinicians should be recommended. Future treatment directions should focus on clear and standardized indications to guide clinicians in the extent of pulp removal to make it easier for dentists to include partial pulpotomy in their everyday practice.

Minimally invasive dental techniques such as partial pulpotomy should also be taught in dental schools and among specialists in pediatric dentistry and endodontics [50,51].

Anyhow the evidence of the interventions under investigation in the present review is not strong enough to support any clinical recommendations. One of the limitations of the performed meta-analysis is that in the clinical trials included, it was impossible to blind the operators to the material used. Moreover, the low number of studies included reduces the level of evidence. This difficulty in finding eligible studies was mainly related to methodological discrepancies in sample/teeth selection and the absence of a precise definition of partial pulpotomy. In particular, some studies were not included in the current review as they included patients with a wide age range and did not provide results according to age. Indeed, some articles were excluded because there was no clear distinction between partial pulpotomy, direct pulp capping, and complete pulpotomy. Another weakness of the study is the aim of extracting as much information as possible about PP studies without a clear distinction of the age groups that were included but not used for the meta-analysis. In addition, there is a lack of data assessing how the success of the therapy affects the quality of life, which can be significantly impaired by pain or functional limitations in chewing. As previously suggested [12], future research should involve rigorously defined methodology and standardization of specific criteria for the techniques used and the variables considered.

Future study directions should include more than just data on materials and techniques. Still, they should also focus on subjects in different age groups to understand what and if there is a limit beyond which PP is not recommended. Furthermore, given the high prevalence of MIH in the paediatric population, studies evaluating the effectiveness of the technique according to the severity of the defect could profoundly influence practitioners’ choices. Moreover, long-term studies could increase the evidence of its effectiveness over time.

In a general scenario in which minimal invasive dentistry is encouraged by scientific evidence, the present review, and others on a similar topic, underline that choosing conservative treatments should be preferable. Given the reduced evidence recorded, eminent guidelines such as those of the American Association of Endodontics or the European Society of Endodontology might be encouraged [48,49].

## 5. Conclusions

The present systematic literature review demonstrates that partial pulpotomy is a successful dental procedure when vital pulpal therapy is required in highly damaged young permanent teeth. Comparable clinical and radiological success was found for all materials used, i.e., mineral trioxide aggregate, MTA + laser, calcium hydroxide, and Biodentine. The biological sparing of partial pulpotomy is an important result that allows more invasive endodontic treatments to be postponed.

## Figures and Tables

**Figure 1 children-10-01447-f001:**
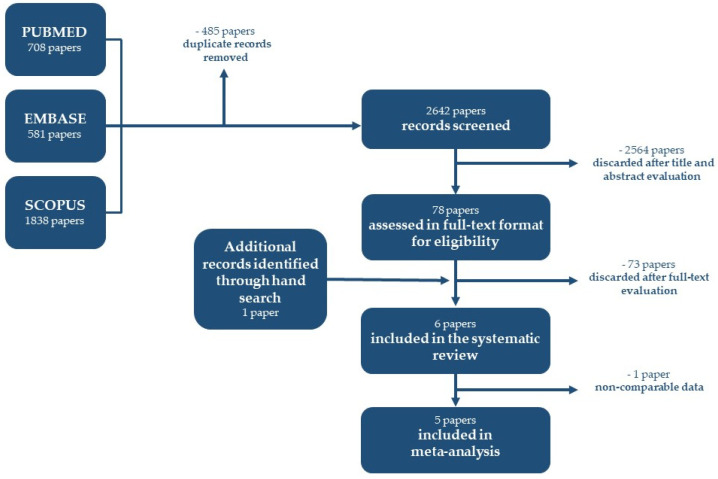
Prisma flowchart.

**Figure 2 children-10-01447-f002:**
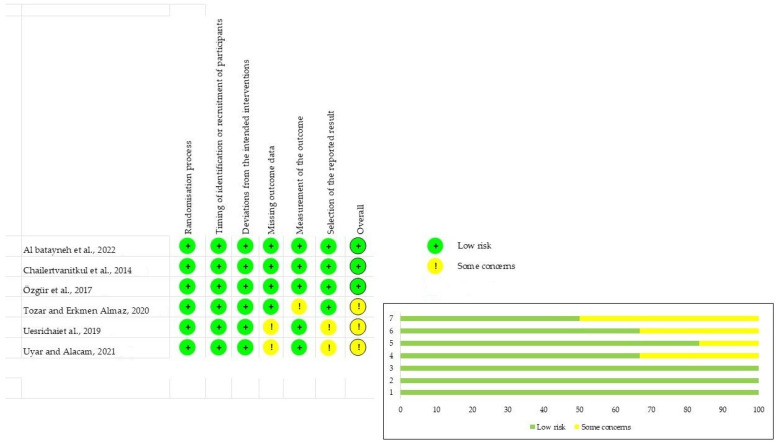
Risk of bias assessment [3,6,33,34,35,36].

**Figure 3 children-10-01447-f003:**
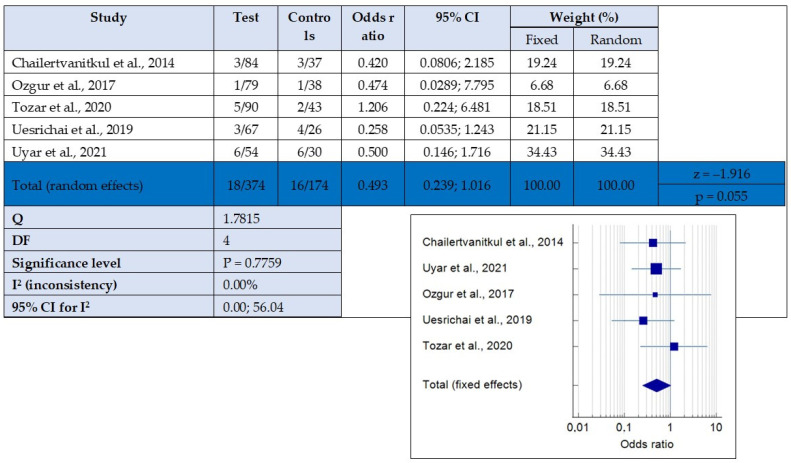
Forest plot of success of partial pulpotomy using mineral trioxide aggregate (MTA) vs. calcium hydroxide, Biodentine, and MTA + laser [3,33,34,35,36].

**Table 1 children-10-01447-t001:** General characteristics of the included studies.

First Author (Year)	Source	Location	Funding Source
Al-batayneh et al., 2022 [6]	*Eur Arch Paed Dent*	Jordan	University grant
Chailertvanitkul et al., 2014 [3]	*Int Endod J*	Thailand	Not reported
Özgür et al., 2017 [37]	*BMC Oral Health*	Turkey	University grant
Tozar and Erkmen Almaz, 2020 [38]	*J Endod*	Turkey	University grant
Uesrichai et al., 2019 [39]	*Int Endod J*	Thailand	University grant
Uyar and Alacam, 2021 [40]	*Niger J Clin Pract*	Turkey	None

**Table 2 children-10-01447-t002:** Main characteristics of the included studies.

Study	Age (Range or Mean)	Patients (*n*)	Teeth (*n*)	Drop-Out of Teeth *n* (%)	Study Design	Final Restoration	Follow-Up (mo.)	Overall Success Rate at 12 mo. (n/Total)		Conclusion
								Test	Control	
Al-batayneh et al., 2022 [6] *	11 ± 3.2 y	50	50	3/50 (6.0%)	PP vs. IPC or FP	GIC, SSC	3, 6, 12, 24	PP 11/12	IPC 2/25	VTP is a valid option in severe carious lesions of permanent first molars with MIH over 24 mo. IPC had a higher success rate (95.8%) than PP or CP (85.7% for both).
									FP 2/11	
Chailertvanitkul et al., 2014 [3]	7–10 y	80	84	8/84 (9.5%)	MTA vs. CH	Amalgam	3, 6, 12, 24	MTA 41/44	CH 37/40	PP using MTA or CH resulted in favourable treatment outcomes. Unfavorable outcomes increase for pulp exposure >5 mm
Özgür et al., 2017 [34]	6–13 y	63	80	4/80 (5.0%)	MTA vs. CH	CR	6, 12, 18, 24	MTA/SH 19/20	CH/SH 19/20	PP with MTA or CH produces comparable and favorable results in immature permanent teeth.
								MTA/SS 20/20	CH/SS 19/19	
Tozar and Erkmen Almaz, 2020 [36]	6–15 y	90	90	3/90 (3.3%)	MTA vs. MTA + Laser Er, Cr: YSGG	GIC, CR	1, 3, 6, 12	MTA 40/45	MTA + Laser 43/45	The use of the laser did not contribute to the success rate compared to MTA alone.
Uesrichai et al., 2019 [35]	10 y	69	69	2/67 (3.0%)	MTA vs. BD	C, SSC	32.2 ± 17.9	MTA 34/37	BD 26/30	Permanent teeth with signs and symptoms of irreversible pulpitis were successfully treated with PP using MTA and BD.
Uyar and Alacam, 2021 [33]	7.9 y	105	119	0/54 (0.0%)	MTA vs. CH or BD	SSC	3, 6, 12	MTA 17/18	CH 13/18	PP treatment is a good option and has high success rates at 12 months with CH, MTA, and BD.
									BD 17/18	

n: number; y: years; mo.: months; PP: partial pulpotomy; CP: cervical pulpotomy; FP: full pulpotomy; VPT: vital pulp therapy; MTA: mineral trioxide aggregate; BD: Biodentine; CH: calcium hydroxide; CR: composite resin; GIC: glass ionomer cement; SSC: stainless steel crown; DCP: direct pulp capping; IPC: indirect pulp capping; SH: sodium hypochlorite; SS: sterile saline. * Not included in meta-analysis.

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
