# Peer review of "Partial Pulpotomy in Young Permanent Teeth: A Systematic Review and Meta-Analysis"

_children, 2023, doi:10.3390/children10091447_

Round 1

Reviewer 1 Report

Abstract

The abstract should follow the style of structured abstracts, but without headings.

Please add abbreviation following full term when possible. The first time an abbreviation appears, it should be placed in parentheses following the full spelling of the term.

Method: Was there a search range in years.

Why other data bases like ScienceDirect, Wiley and Cochrane were not used.

Systematic review registration should be mentioned after the abstract and before keywords. Please check journal guidelines.

Keywords need to follow the MESH.Introduction

The Introduction does not give a rationale why this study should be conducted or what scientific value it has.

At the end of the Intro section, please give your null hypothesis. The latter should be derived from the preceding thoughts in this section and should be broached again in the Discussion. In hypothesis testing, the null hypothesis is the one you are hoping that is can be disproven by the observed data.

Please add ref number after the author name/s.

Please add subtitle: Rationale of study and objective.

Materials and Methods

What is the study rationale?

Why other data bases like ScienceDirect, Wiley and Cochrane were not used.

Please add subtitle: Quality Assessment

Please add abbreviation following full term when possible. The first time an abbreviation appears, it should be placed in parentheses following the full spelling of the term.

Please add the name of the manufacturer for all materials, X-rays and equipment used.

Results

Please add subtitle: Reliability and Validity.

Please add P-values.

Tables and Figures

In all Tables, a footnote explaining the abbreviations needs to be added (what do they stands for). Also, level of significance needs to be mentioned.

Discussion

This section may usefully start with a summary of the major findings, but repetition of parts of the abstract or of the results section should be avoided.

“In the present/current review” NOT “In this/our review”.

Please add ref number after the author name/s.

Please mention future directions.

References

References needs to be 10 years back not more (from 2013 to 2023).

Old references need to be replaced by recent ones.

Some references have missing information. Some of the references include DOI, others do not include DOI number.

In general, all references need to be revised, standardized, and written according to the journal guidelines. Some references have missing information. Also, some of the journals were abbreviated, while others were written in full term.

Moderate editing of English language required

Author Response

We would like to thank the reviewer for taking the time to review the paper.

Abstract

  • The abstract should follow the style of structured abstracts, but without headings.

It was modified.

  • Please add abbreviation following full term when possible.The first time an abbreviation appears, it should be placed in parentheses following the full spelling of the term.

Abbreviations were corrected.

  • Method: Was there a search range in years.

We  have corrected it, specifying the date of the search.

  • Why other data bases like ScienceDirect, Wiley and Cochranewere not used.

We decided to use three of the most widely used databases, in addition to the grey literature search with Google Scholar. According to a recent article, searching in at least two databases already reduces the risk of missing relevant studies. (Ewald H, Klerings I, Wagner G, Heise TL, Stratil JM, Lhachimi SK, Hemkens LG, Gartlehner G, Armijo-Olivo S, Nussbaumer-Streit B. Searching two or more databases decreased the risk of missing relevant studies: a metaresearch study. J Clin Epidemiol. 2022 Sep;149:154-164. doi: 10.1016/j.jclinepi.2022.05.022. Epub 2022 May 30. PMID: 35654269.)

  • Systematic review registration should be mentioned after the abstract and before keywords. Please check journal guidelines.

It was reported as suggested.

  • Keywords need to follow the MESH.

Now all the keywords are Mesh terms.

Introduction

The Introduction does not give a rationale why this study should be conducted or what scientific value it has.

  • At the end of the Intro section, please give your null hypothesis. The latter should be derived from the preceding thoughts in this section and should be broached again in the Discussion. In hypothesis testing, the null hypothesis is the one you are hoping that is can be disproven by the observed data.

It was added

  • Please add ref number after the author name/s.

It was added

  • Please add subtitle: Rationale of study and objective.

It was added

Materials and Methods

  • What is the study rationale?

We add the rationale in the introduction as previously required

  • Why other data bases like ScienceDirect, Wiley and Cochranewere not used.

We decided to use three of the most widely used databases and, to extend the search, the grey literature was searched using Google Scholar. In fact, according to a recent article, searching in at least two databases already decreases the risk of missing relevant studies. (Ewald H, Klerings I, Wagner G, Heise TL, Stratil JM, Lhachimi SK, Hemkens LG, Gartlehner G, Armijo-Olivo S, Nussbaumer-Streit B. Searching two or more databases decreased the risk of missing relevant studies: a metaresearch study. J Clin Epidemiol. 2022 Sep;149:154-164. doi: 10.1016/j.jclinepi.2022.05.022. Epub 2022 May 30. PMID: 35654269.)

  • Please add subtitle: Quality Assessment

It was added

  • Please add abbreviation following full term when possible.The first time an abbreviation appears, it should be placed in parentheses following the full spelling of the term.

They were added.

  • Please add the name of the manufacturer for all materials, X-rays and equipment used.

The name of manufacturer was added, while for the x-rays we specified that all the studies used intraoral radiographs (standard type of x-rays for single tooth evaluation) as no further details were given.

Results

  • Please add subtitle: Reliability and Validity.

It was added

  • Please add P-values.

P-values are mentioned in the quantitative synthesis and in Figure 3

Tables and Figures

In all Tables, a footnote explaining the abbreviations needs to be added (what do they stands for). Also, level of significance needs to be mentioned.

Now, you can find the abbreviations and the level of significance in figure 3

Discussion

This section may usefully start with a summary of the major findings, but repetition of parts of the abstract or of the results section should be avoided.

The start of the discussion was modified, according to the reviewer’s suggestion

  • “In the present/current review” NOT “In this/our review”.

Corrections were made.

  • Please add ref number after the author name/s.

Done

  • Please mention future directions.

The following sentence was added: “Future study directions should include more than just data on materials and techniques. Still, they should also focus on subjects in different age groups to understand what and if there is a limit beyond which PP is not recommended. Furthermore, given the high prevalence of MIH in the paediatric population, studies evaluating the effectiveness of the technique according to the severity of the defect could profoundly influence practitioners' choices. Moreover, long term studies. Moreover, long-term studies could increase the evidence of its effectiveness over time.”

References

References needs to be 10 years back not more (from 2013 to 2023).

  • Old references need to be replaced by recent ones.

References were replaced.

  • Some references have missing information. Some of the references include DOI, others do not include DOI number.

We add DOI, where possible

  • In general, all references need to be revised, standardized, and written according to the journal guidelines. Some references have missing information. Also, some of the journals were abbreviated, while others were written in full term.

We apologize for the wrong format of the references; we reviewed them using Endnote and when information was missing, manual editing was done.

Reviewer 2 Report

Thank you for allowing me to review this systematic review, the purpose of which was to evaluate the clinical and radiographic success of partial pulpotomies in deep caries processes or post-eruptive defects in young, vital permanent teeth.

The review paper has a well-defined research question and database selection is explained. The inclusion and exclusion criteria are also listed, the literature selection is up-to-date and the manuscript include overview tables characterizing each study in the sample.

It is necessary to make certain corrections:

1. Remove numbers from keywords.

2. Please provide implications for (future) research and practice.

Author Response

We would like to thank the Reviewer for his/her time and appreciation of our work. The numbers from the keywords were removed and the following sentence was added at the end of the discussion “Future study directions should include more than just data on materials and techniques. Still, they should also focus on subjects in different age groups to understand what and if there is a limit beyond which PP is not recommended. Furthermore, given the high prevalence of MIH in the paediatric population, studies evaluating the effectiveness of the technique according to the severity of the defect could profoundly influence practitioners' choices. Moreover, long term studies. Moreover, long-term studies could increase the evidence of its effectiveness over time.”

Round 2

Reviewer 1 Report

None